# Analysis of *Neisseria gonorrhoeae* and *Mycoplasma genitalium* from nucleic acid amplification test specimens, Nunavut region of Inuit Nunangat, Canada, 2020–2023

Norman Barairo,[1] Ameeta E. Singh,[2] Shelley Peterson,[1] Ekua Agyemang,[3] Irene Martin[1]

**ABSTRACT**  This study analyzed *Neisseria gonorrhoeae* (GC) and *Mycoplasma genitalium* (Mgen) from remnant nucleic acid amplification test (NAAT) specimens in Nunavut, Canada (2020–2023) to assess strain distribution, antimicrobial resistance (AMR), and co-infection rates. Among 2,221 GC-positive specimens, no cephalosporin resistance was detected, but intermediate-to-decreased susceptibility increased in 2023. Predicted ciprofloxacin resistance was 45%, and azithromycin resistance remained stable at 9.6%. Of 778 tested specimens, 31.4% were co-infected with Mgen, with 49.2% associated with azithromycin resistance, but none with moxifloxacin resistance. These findings highlight emerging AMR trends and support reassessment of current treatment guidelines. Remnant urine specimens submitted for nucleic acid amplification testing identified differences in the prevalence and AMR patterns in GC and Mgen in a large Canadian Arctic territory.

**IMPORTANCE**  Nunavut has the highest gonorrhea rates in Canada, yet antimicrobial resistance (AMR) surveillance is limited to culture-based testing. This study bridges that gap by using molecular assays to predict AMR directly from remnant nucleic acid amplification test specimens, providing data for remote regions that lack the resources to culture and transport *Neisseria gonorrhoeae* cultures. Findings highlight relevant AMR trends for gonorrhea treatment. Mgen, despite increasing clinical prevalence and high co-infection rates with other sexually transmitted infections, is not included in national surveillance programs, furthering the need for extended monitoring.

**KEYWORDS**  gonorrhea, *Mycoplasma genitalium*, antimicrobial resistance

*N*eisseria gonorrhoeae (GC) is the causative agent of gonorrhea—the second most reported sexually transmitted infection (STI) in Canada. Nunavut has the highest GC incidence nationwide (1,420.3 cases per 100,000 in 2021)—over 16 times the national average (1). Nunavut, one of the four regions in Inuit Nunangat—the homeland of the Inuit—is separated into three regions: Kitikmeot (west), Kivalliq (center), and Qikiqtaaluk (east) (2, 3).

The Gonococcal Antimicrobial Surveillance Program of Canada (GASP-Canada) characterizes and monitors antimicrobial resistance (AMR) in gonorrhea through phenotypic testing and whole genome sequencing, which are limited to viable cultures (4). Approximately 10% of gonorrhea cases are diagnosed by cultures, with 90% diagnosed by nucleic acid amplification tests (NAATs) (1). Culturing and transporting viable GC cultures from remote regions is challenging, creating a gap in AMR monitoring. To address this, we used molecular assays for sequence typing and AMR predictions from NAAT specimens.

Address correspondence to Ameeta E. Singh, ameeta@ualberta.ca.

The authors declare no conflicts of interest

*Mycoplasma genitalium* (Mgen), an emerging STI, is associated with non-gonococcal urethritis and cervicitis and is often co-infected with other STIs (5). Mgen testing is not routinely available in Canada, limiting prevalence data. Resistance to azithromycin and moxifloxacin is of concern, as these are the current recommended treatment options (5, 6).

## MATERIALS AND METHODS

We tested 2,221 GC NAAT-positive urine specimens collected in Nunavut (2020–2023): 216 (9.7%) from Kitikmeot, 675 (30.4%) from Kivalliq, and 1,330 (59.9%) from Qikiqtaaluk (Table S1). These specimens underwent nucleic acid extractions and PCR for *Neisseria gonorrhoeae* Multi-Antigen Sequence Typing (NG-MAST) as previously described (7, 8). Single nucleotide polymorphism (SNP) detection assays were conducted using real-time PCR to predict AMR to cephalosporins (ceftriaxone, cefixime), ciprofloxacin, and azithromycin (7, 9). For cephalosporins, specimens were predicted resistant if *penA* A311V was detected; intermediate-to-decreased susceptible (I-DS) if at least three SNPs of *ponA* L421P, *mtrR* delA, *porB* G120/A121, or *penA* A501/N513Y/G543S were detected. Detection of any SNP associated with resistance to ciprofloxacin (*gyrA* S91, *parC* D86/S87/S88) or azithromycin (*mtrR* promoter, 23S rRNA A2059G/C2611T) led to predicted resistance for that antimicrobial.

GC strains with the same NG-MAST ST have been shown to have similar AMR profiles (10); therefore, AMR prediction testing was performed on a maximum of 30 samples for each ST, and the remaining results were inferred if the same AMR interpretation was predicted for each antimicrobial (3).

Specimens from 2020 to 2021 with sufficient remaining sample volume ($n = 778$) were also tested for Mgen presence and AMR to azithromycin and moxifloxacin. Detection of Mgen was done via real-time PCR targeting the *MgPa* adhesin protein (5). For Mgen AMR, PCR, and Sanger sequencing were used to detect SNPs in the 23S rRNA gene (A2058G/C, A2059C/G/T) for azithromycin resistance, and in the *gyrA* (M95I, D998A/Y, and F108I/Y) and *parC* (S83I/R/N/C, D87Y/V/G/N, and L97R) genes for moxifloxacin resistance (5, 6, 11).

## RESULTS

Of 2,221 specimens, 245 (11.0%) were nontypeable for NG-MAST due to low nucleic acid concentrations. Among the remaining 1,976 specimens, 184 different NG-MAST STs were detected. ST-1993 ($n = 407$, 20.7%) and ST-20400 ($n = 231$, 11.7%) were the most prevalent (Fig. S1). ST-1993 was primarily found in Qikiqtaaluk, whereas ST-20400 was the most common in Kitikmeot and Kivalliq.

No resistance to cephalosporins was detected, though 19.6% of cases had I-DS, highest in Kitikmeot (49.1%). Predicted ciprofloxacin resistance was 45.0% overall, also highest in Kitikmeot (58.8%). Predicted azithromycin resistance was 9.6% overall, highest in Qikiqtaaluk (11.4%) (Fig. 2, Table S2).

Mgen was detected in 31.4% (224/778) of samples tested (Table 1). Kivalliq had the highest rate (39.0%), more than double Kitikmeot (16.1%). Among the 244 Mgen infections, none had mutations associated with moxifloxacin resistance, and 49.2% ($n = 120$) had predicted azithromycin resistance, of which 9.2% were also associated with gonococcal azithromycin resistance. More than half of Mgen-positive samples tested in

**TABLE 1** Rates of predicted azithromycin resistance in *Mycoplasma genitalium* and *Neisseria gonorrhoeae* (GC) of NAAT samples from Nunavut, Canada (2020–2021)

| Region | GC-positive NAAT specimens tested for Mgen (2020–2021), *n* (%) | Mgen positive, *n* (%) | Azithromycin-resistant Mgen, *n* (%) | Azithromycin-resistant GC and Mgen, *n* (%) |
|---|---|---|---|---|
| Kitikmeot | 31 (4.0%) | 5/31 (16.1) | 2/5 (40.0) | 1/2 (50.0) |
| Kivalliq | 177 (22.8%) | 69/177 (39.0) | 31/69 (44.9) | 3/31 (9.7) |
| Qikiqtaaluk | 570 (73.3%) | 170/570 (29.8) | 87/170 (51.2) | 7/87 (8.1) |
| Overall | 778 | 244/778 (31.4) | 120/244 (49.2) | 11/120 (9.2) |

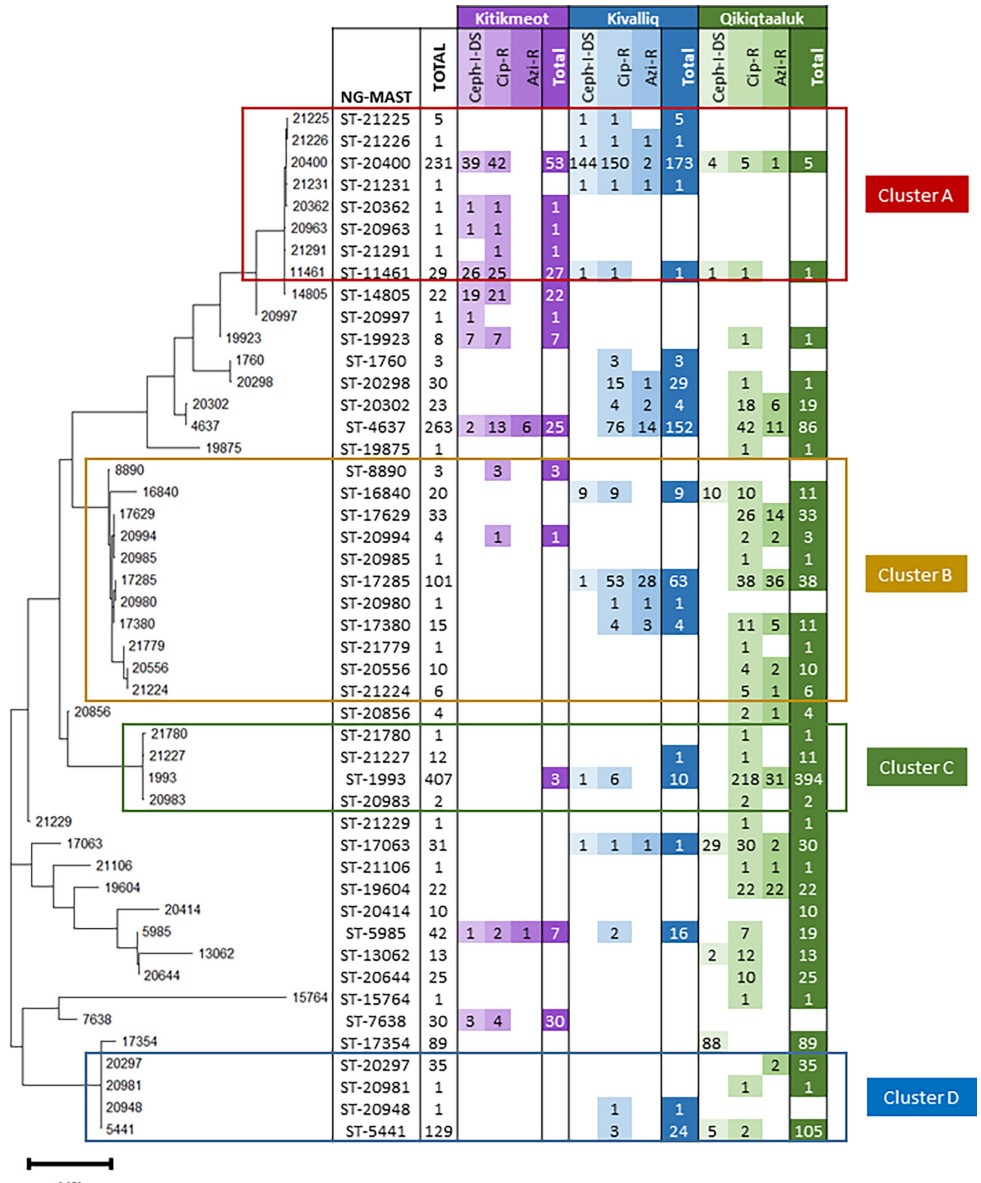

**FIG 1** Phylogenetic tree of *Neisseria gonorrhoeae*-positive Nunavut NAAT NG-MAST sequence types from Nunavut, Canada, 2020–2023 (12, 13). The most prevalent STs and those associated with AMR are included in this figure.

Qikiqtaaluk were associated with azithromycin resistance (51.2%), with slightly lower rates in Kivalliq (44.9%) and Kitikmeot (40%) (Table 1).

## DISCUSSION

The identified NG-MAST STs were classified into four distinct phylogenetic clusters (Fig. 1). Cluster A largely encompassed STs from Kitikmeot that were associated with I-DS to cephalosporins and ciprofloxacin resistance. Cluster D largely comprised STs from Qikiqtaaluk that were mostly susceptible.

Predicted ciprofloxacin resistance increased from 21.5% in 2020 to 52.7% in 2021, stabilizing thereafter, and was comparable to GASP-Canada's rate (58.7%, 2022) (4) (Fig. 2). Predicted I-DS (≥0.032 mg/L) to cephalosporins rose from 9.2% (2022) to 38.0% (2023), which was linked with increasing ST-20400 prevalence. This led to I-DS rates in 2023 surpassing GASP-Canada's rates (2022) (4), which were 6.5% for ceftriaxone (221/3,393)

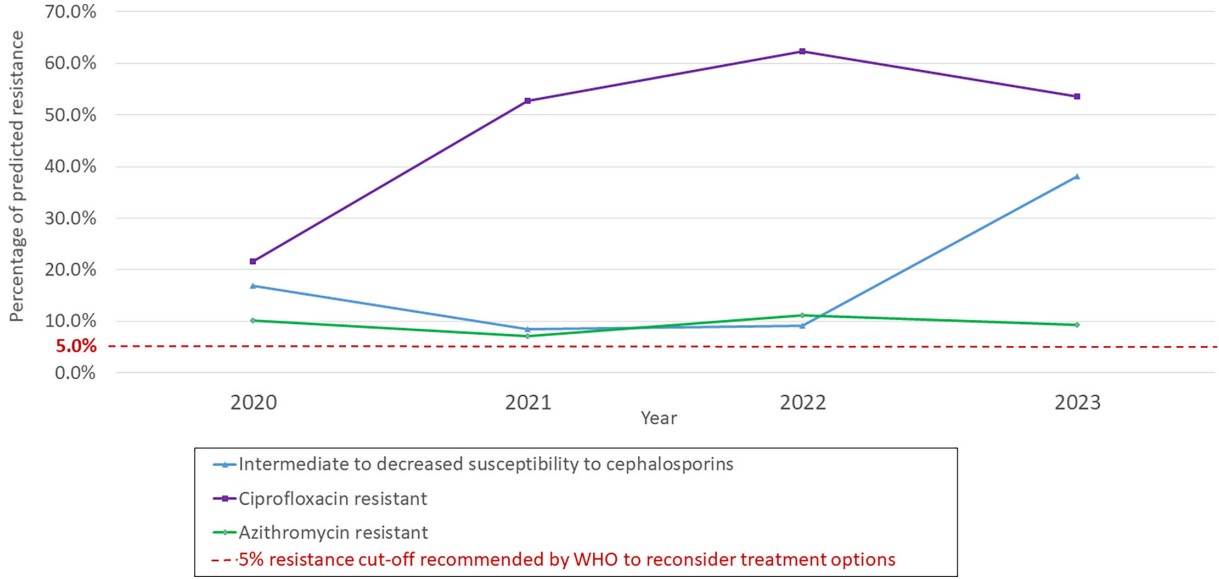

**FIG 2** *Neisseria gonorrhoeae* antimicrobial resistance prediction rates from *Neisseria gonorrhoeae*-positive NAAT specimens from Nunavut, Canada, 2020–2023.

and 19.9% for cefixime (676/3,393) (8). Predicted azithromycin resistance ranged from 7.0% to 11.1% and was lower than GASP-Canada's rate, 2022 (≥1 mg/L) (33.4%) (3).

Mgen co-infection rates varied geographically, with Kivalliq having the highest (39%) and Kitikmeot the lowest (16.1%). The rates of macrolide resistance ranged from 40% to 51% of Mgen-positive specimens, which is lower than rates reported elsewhere in Canada (56.5%, 63.6%), as well as the global estimate of >50% (14–16). The absence of fluoroquinolone resistance in Mgen specimens is of particular interest, as a previous Canadian study reported 12.2% resistance, and the estimated global fluoroquinolone resistance rate is 7.7% (14, 16).

Only urine specimens were available for testing, limiting representation across anatomical sites. Molecular AMR predictions may overestimate resistance, as genetic mutations do not always confer phenotypic resistance. Additionally, Mgen was only tested on GC-positive samples, limiting the assessment of overall Mgen prevalence. Despite these limitations, findings provide critical insights into AMR trends in Nunavut.

Recent Nunavut gonorrhea treatment guidelines recommend ceftriaxone/cefixime plus azithromycin. Removal of azithromycin from routine treatments for gonorrhea may help to mitigate further progression of macrolide resistance in both gonorrhea and Mgen (17). Given the high prevalence of azithromycin resistance in GC/Mgen co-infections (9.6% GC, 49.2% Mgen) (Table S2), continued monitoring of AMR trends in both pathogens is essential for informed treatment recommendations.

## Conclusions

NG-MAST and SNP assays provided insights into GC strain distribution and AMR in Nunavut. No cephalosporin resistance was detected, but rising I-DS to cephalosporins in 2023 warrants continued monitoring. Predicted azithromycin resistance rates exceeded the WHO's 5% threshold for treatment reconsideration (18).

Approximately 30% of GC-positive samples were coinfected with Mgen. Nearly half of Mgen samples were resistant to azithromycin, the first-line treatment; however, moxifloxacin remains a valid treatment option, as no resistance was detected. Continued monitoring of strains and AMR is essential to guide management of GC and Mgen in Nunavut.

## ACKNOWLEDGMENTS

The authors are grateful to the staff, including Kim Dionne at the Qikiqtani General Hospital Lab, for coordinating testing, aliquoting, packing, and shipping of the positive specimens used for the study and to Kethika Kulleperuma and Serges Kabore for epidemiological support. We thank Nick Nordal-Budinsky, Giulia Severini, and Paige Adams from the Streptococcus and STI Section, National Microbiology Laboratory, for their technical assistance on this study. We acknowledge the biomedical and technical laboratory approach of this work and the limited partnership or knowledge sharing elements with Nunavut communities. Efforts are underway to improve knowledge sharing through sexual health programs in Nunavut.

This project was supported by internal funding from the Public Health Agency of Canada.

## AUTHOR AFFILIATIONS

[1]National Microbiology Laboratory Branch, Public Health Agency of Canada, Winnipeg, Manitoba, Canada
[2]Division of Infectious Diseases, University of Alberta, Edmonton, Alberta, Canada
[3]Government of Nunavut, Iqaluit, Nunavut, Canada

## AUTHOR ORCIDs

Norman Barairo http://orcid.org/0009-0000-4260-5225
Ameeta E. Singh http://orcid.org/0000-0003-2627-3704
Shelley Peterson http://orcid.org/0000-0002-6432-9257
Irene Martin http://orcid.org/0000-0002-3941-5583

## AUTHOR CONTRIBUTIONS

Norman Barairo, Data curation, Formal analysis, Investigation, Methodology, Software, Validation, Writing – original draft | Ameeta E. Singh, Conceptualization, Writing – review and editing | Shelley Peterson, Conceptualization, Data curation, Formal analysis, Investigation, Methodology, Project administration, Supervision, Writing – original draft, Writing – review and editing | Ekua Agyemang, Supervision, Writing – review and editing | Irene Martin, Conceptualization, Data curation, Funding acquisition, Methodology, Project administration, Resources, Supervision, Writing – original draft, Writing – review and editing

## ETHICS APPROVAL

Ethics approval for conducting this study was obtained from the University of Alberta Health Research Ethics Board (Pro00119186) and the Nunavut Research Institute (Scientific Research License 05 011 22R-M).

## ADDITIONAL FILES

The following material is available online.

### Supplemental Material

**Supplemental tables and figure (Spectrum01553-25-S0001.docx).** Tables S1 and S2, and Fig. S1.

### Open Peer Review

**PEER REVIEW HISTORY (review-history.pdf).** An accounting of the reviewer comments and feedback.

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
