## [Reviewer comments · Microbiology Spectrum]

Microbiology Spectrum

Analysis of *Neisseria gonorrhoeae* and *Mycoplasma genitalium* from Nucleic Acid Amplification Test Specimens, Nunavut Region of Inuit Nunangat, Canada, 2020-2023

Norman Barairo, Ameeta Singh, Shelley Peterson, Ekua Agyemang, and Irene Martin

Corresponding Author(s): Ameeta Singh, University of Alberta

Review Timeline:

Submission Date:	May 18, 2025
Editorial Decision:	July 21, 2025
Revision Received:	July 24, 2025
Accepted:	July 25, 2025

Editor: Ayush Kumar

Reviewer(s): The reviewers have opted to remain anonymous.

Transaction Report:

DOI: <https://doi.org/10.1128/spectrum.01553-25>

Re: Spectrum01553-25 (**Analysis of *Neisseria gonorrhoeae* and *Mycoplasma genitalium* from Nucleic Acid Amplification Test Specimens, Nunavut Region of Inuit Nunangat, Canada, 2020-2023**)

Dear Dr. Ameeta E Singh:

Thank you for the privilege of reviewing your work. Below you will find my comments, instructions from the Spectrum editorial office, and the reviewer comments.

I have also carefully reviewed the manuscript and agree with the reviewer's comment that the two figures require further clarification. I am also interested in the data sharing arrangements with the participating communities. Were the data shared with each of the communities included in the study? Additionally, do the authors require any approvals, in accordance with OCAP principles, prior to publishing these data? I would prefer that this is explicitly mentioned in the manuscript.

Revision Guidelines

Sincerely,
Ayush Kumar
Editor
Microbiology Spectrum

This paper examines the presence of *M. genitalium* and *N. gonorrhoeae* in urine samples over 4 years from the Nunavut province in Canada. It draws awareness to the high STI rates but limitations of culture testing in remote Canada and demonstrates that molecular assays are a good substitute for predicting AMR when culture is not an option. The results are very important for knowing how STIs in more remote locations can be diagnosed and appropriately treated given the geographical limitations. It also shows how their trends fit or are different, into global and national population trends adding to the collective knowledge on Gc and Mg infections and co-infections.

I found this paper concise and well written; it was clear to understand and interpret. The discussion acknowledged its limitations being only urine specimens and as such did not make any overreaching conclusions.

My only comments for improvement are on the graphs-

Figure 2 is missing most of its key (orange and green) and the axes need titles. What does the red dotted line show?

Figure S1 is also missing part of the key (green) and its axis titles.

Response to Reviewer's and Editor's comments on **Analysis of *Neisseria gonorrhoeae* and *Mycoplasma genitalium* from Nucleic Acid Amplification Tests, Nunavut Region of Inuit Nunangat, Canada, 2020-2023**

Reviewer's comments	Response
Figure 2 is missing most of its key (orange and green) and the axes need titles. What does the red dotted line show?	Changes have been made to Figure 2 to address these comments
Figure S1 is also missing part of the key (green) and its axis titles.	Changes have been made to Figure S1 to address these comments
Editor's comments	Response
I am also interested in the data sharing arrangements with the participating communities. Were the data shared with each of the communities included in the study? Additionally, do the authors require any approvals, in accordance with OCAP principles, prior to publishing these data? I would prefer that this is explicitly mentioned in the manuscript.	Due to the relatively small numbers of samples and large number of communities, community level data was not shared with each community Approval was obtained from the University of Alberta and the Nunavut Research Institute (NRI) prior to conducting this study; this has now been added to the Methods section. In addition, approval for publication was obtained from Dr. Agyemang (CMOH), who is also listed as an author on this study.

Re: Spectrum01553-25R1 (**Analysis of *Neisseria gonorrhoeae* and *Mycoplasma genitalium* from Nucleic Acid Amplification Test Specimens, Nunavut Region of Inuit Nunangat, Canada, 2020-2023**)

Dear Dr. Ameeta E Singh:

Your manuscript has been accepted, and I am forwarding it to the ASM production staff for publication. Your paper will first be checked to make sure all elements meet the technical requirements. ASM staff will contact you if anything needs to be revised before copyediting and production can begin. Otherwise, you will be notified when your proofs are ready to be viewed.

Sincerely,
Ayush Kumar
Editor
Microbiology Spectrum